# RTCB Complex Regulates Stress-Induced tRNA Cleavage

**DOI:** 10.3390/ijms232113100

**Published:** 2022-10-28

**Authors:** Yasutoshi Akiyama, Yoshika Takenaka, Tomoko Kasahara, Takaaki Abe, Yoshihisa Tomioka, Pavel Ivanov

**Affiliations:** 1Laboratory of Oncology, Pharmacy Practice and Sciences, Tohoku University Graduate School of Pharmaceutical Sciences, Sendai 980-8578, Japan; 2Department of Clinical Biology and Hormonal Regulation, Tohoku University Graduate School of Medicine, Sendai 980-8574, Japan; 3Department of Medical Science, Tohoku University Graduate School of Biomedical Engineering, Sendai 980-8574, Japan; 4Division of Rheumatology, Inflammation and Immunity, Brigham and Women’s Hospital, Boston, MA 02115, USA; 5Department of Medicine, Harvard Medical School, Boston, MA 02115, USA

**Keywords:** tRNA, RTCB ligase complex, oxidative stress, tiRNAs, angiogenin

## Abstract

Under stress conditions, transfer RNAs (tRNAs) are cleaved by stress-responsive RNases such as angiogenin, generating tRNA-derived RNAs called tiRNAs. As tiRNAs contribute to cytoprotection through inhibition of translation and prevention of apoptosis, the regulation of tiRNA production is critical for cellular stress response. Here, we show that RTCB ligase complex (RTCB-LC), an RNA ligase complex involved in endoplasmic reticulum (ER) stress response and precursor tRNA splicing, negatively regulates stress-induced tiRNA production. Knockdown of RTCB significantly increased stress-induced tiRNA production, suggesting that RTCB-LC negatively regulates tiRNA production. Gel-purified tiRNAs were repaired to full-length tRNAs by RtcB in vitro, suggesting that RTCB-LC can generate full length tRNAs from tiRNAs. As RTCB-LC is inhibited under oxidative stress, we further investigated whether tiRNA production is promoted through the inhibition of RTCB-LC under oxidative stress. Although hydrogen peroxide (H_2_O_2_) itself did not induce tiRNA production, it rapidly boosted tiRNA production under the condition where stress-responsive RNases are activated. We propose a model of stress-induced tiRNA production consisting of two factors, a trigger and booster. This RTCB-LC-mediated boosting mechanism may contribute to the effective stress response in the cell.

## 1. Introduction

When cells encounter various stress conditions, they trigger stress response pathways to protect themselves. As protein synthesis is an energy-expensive process, under stress conditions, cells generally repress translation to conserve energy and nutrients for recovery. Stress-induced translation repression is generally mediated by two major stress-responsive pathways, the integrated stress response (ISR) pathway and the mechanistic target of rapamycin complex 1 (mTORC1) pathway (reviewed in [1]). In addition, accumulating evidence shows that the metabolism of transfer RNAs (tRNAs) also plays important roles in translation repression as a part of cellular stress response.

Besides the canonical role in delivering amino acids to the ribosome, tRNAs are cleaved into fragments by specific ribonucleases (RNases) in response to stress stimuli, generating tRNA-derived RNAs that contribute to various aspects of cellular physiology. One of the key RNases for such stress responses is angiogenin (ANG), a member of the vertebrate-specific RNase A superfamily enzymes (reviewed in [2,3]). Under non-stress conditions, ANG is located in the nucleus, or resides in the cytoplasm as a complex with RNH1, an endogenous inhibitor of RNase A superfamily enzymes. Under various stress conditions such as oxidative stress, nutrient deprivation, heat shock, and UV irradiation, ANG becomes activated through its dissociation from RNH1 (reviewed in [3]). In addition, nuclear ANG translocates to the cytoplasm [4]. The activated ANG targets the anticodon loop of mature tRNAs for cleavage in the cytoplasm, generating tRNA halves called tRNA-derived stress-induced RNAs (tiRNAs) [5,6]. Among them, only 5′-tiRNAs showed the activity of translation repression [5]. Further characterization of individual 5’-tiRNAs revealed that specific 5′-tiRNAs derived from tRNA^Ala^ and tRNA^Cys^ (5′-tiRNA^Ala^ and 5′-tiRNA^Cys^) inhibit translation initiation [7]. The translation inhibition activity of 5’-tiRNA^Ala^ and 5′-tiRNA^Cys^ are mediated by a stretch of five guanosines at their 5′-end, termed the terminal oligoguanine (TOG) motif. Through the TOG motifs, these 5′-tiRNAs form G-quadruplex (G4) structures [8,9,10], stable non-canonical four-stranded RNA structures formed in guanine (G)-rich sequences (reviewed in [11]). TOG motif-mediated G4 assembly is essential for translation inhibition activity of these 5′-tiRNAs. Mechanistically, G4-tiRNAs directly bind to the eukaryotic initiation factor eIF4G, one of the subunits of cap-binding eIF4F complex, displacing the eIF4G from the cap structure of mRNAs, which results in repression of the scanning step of translation initiation [12]. In addition, G4-tiRNAs induce the assembly of stress granules [13,14], membraneless cytoplasmic RNA granules considered to promote survival under stress conditions (reviewed in [15,16,17]). It was also reported that tiRNAs interact with cytochrome *C* (Cyt *C*) released from mitochondria under osmotic stress, preventing apoptosis by inhibiting apoptosome formation through sequestering of Cyt *C* from Apaf-1 [18].

Growing evidence suggest that the regulation of tiRNA production, especially ANG-mediated tiRNA production, is essential for various physiological and pathological processes, such as survival during stress (see above), tumorigenesis and cancer progression [19,20,21,22], neurodegenerative disorders [9], and differentiation of stem cells [23,24,25]. Therefore, it is important to better understand the regulatory mechanisms of tiRNA production. As tiRNAs are generated by RNase-mediated cleavage, the expression levels of intracellular RNases and their availability are the primary determinant of the net tiRNA production. Recently, it has been shown that tiRNAs can be produced in an ANG-independent manner [26,27]. Using RNH1 knockout cells, we showed that the net amount of sodium arsenite-induced tiRNAs is largely dependent on the dissociation of RNH1 from all the RNase A superfamily enzymes (including ANG) expressed in the cell [27]. A recent study suggested that RNase 1, one of RNase A superfamily enzymes, is mainly responsible for ANG-independent tiRNA production [28]. Therefore, both intracellular levels of RNase A superfamily enzymes and stress-induced dissociation of RNH1 from the RNases determine the amount of tiRNA production. In addition, some tRNA modifications also affect the efficiency of ANG-mediated tRNA cleavage (reviewed by [29]). For example, 5-methylcytosine (m^5^C) modifications catalyzed by methyltransferases NSUN2 or DNMT2 protect tRNAs from ANG-mediated cleavage [30,31,32]. Mice lacking both DNMT2 and NSUN2 showed a complete lack of m^5^C modifications in specific tRNA species, and showed reduced overall protein synthesis [30], suggesting that m^5^C modifications are decisive for the regulation of protein synthesis. A recent study showed that NSUN2-mediated m^5^C modifications are inhibited by oxidative stress, resulting in reduced translation rates due to enhancement of ANG-mediated tRNA cleavage [33], which suggests that translation rates are dynamically regulated by NSUN2-mediated m^5^C modifications in response to environmental conditions to maintain cellular homeostasis. Although these factors are essential for the regulation of tiRNA production, it remains unclear whether other critical mechanisms exist to regulate the efficiency of tiRNA production.

In this study, we focused on the RTCB ligase complex as a candidate molecule to regulate stress-induced tiRNA production. Human RTCB ligase complex (RTCB-LC) is an RNA ligase complex comprising of at least four core subunits, RTCB, DDX1, FAM98B and CGI-99 [34]. While RTCB is the only subunit essential for the ligation reaction, it requires Archease (ARCH) as a cofactor for multiple turnover reactions [35]. RTCB was first identified as the essential subunit of a human tRNA splicing ligase complex [36]. Later it was reported that RTCB is also responsible for the ligation of XBP1 mRNA exons after IRE1α-mediated excision of the intron as a part of unfolded protein response (UPR) [37,38,39]. The unique feature of the complex is that it directly joins 2′,3′-cyclic phosphate (cP) residue and 5′-hydroxyl (5′-OH) residue [40,41]. As 5′-tiRNAs and 3′-tiRNAs also have cP and 5′-OH, respectively (reviewed in [42]), we hypothesized that the production of tiRNAs may be regulated by the RTCB-LC. Here, we reveal that the RTCB-LC negatively regulates tiRNA production. We also show that under oxidative stress, tiRNA production is enhanced through oxidative inhibition of the RTCB-LC. We propose a model of the regulation of stress-induced tiRNA production consisting of two factors, a trigger and booster. This RTCB-LC-mediated boosting mechanism may enable cells to trigger tiRNA-mediated stress response rapidly and efficiently.

## 2. Results

### 2.1. RTCB Ligase Complex Negatively Regulates tiRNA Production

As the RTCB-LC generally functions to ligate two RNA fragments into a single product, it is expected that the RTCB-LC acts to decrease the amount of tiRNAs. We first examined whether RTCB knockdown enhanced tiRNA-production under several conditions. We generated four shRNA constructs for RTCB knockdown (Appendix A). While RTCB knockdown itself did not induce the obvious RNA degradation or fragmentation (Appendix A), Northern blotting showed that RTCB knockdown induced the accumulation of pre-tRNA^Tyr^-derived fragments (Appendix A), that were generated due to the failure in the RTCB-mediated ligation of pre-tRNA^Tyr^ exons [43]. It should be noted that the amount of pre-tRNA^Tyr^-derived fragment is in the direct proportion to the RTCB knockdown efficiency, suggesting that RTCB knockdown directly impacts on the enzymatic activity of the RTCB-LC. As the construct #1 showed the strongest knockdown efficiency (Appendix A), we used this shRNA for the subsequent experiments. First, we investigated the effect of RTCB knockdown on ANG-mediated tiRNA production (Figure 1A–C). shRNA-mediated RTCB knockdown significantly decreased the protein levels of RTCB at day 7 after transduction (Figure 1A). Under the condition, we treated U2OS cells with recombinant ANG. Recombinant ANG added to cell culture is rapidly internalized and cleaves cytoplasmic tRNAs into tiRNAs [5]. While RTCB knockdown did not induce any tiRNA production, it significantly enhanced ANG-mediated tiRNA production (Figure 1B,C and Appendix A), suggesting that the RTCB-LC negatively regulates tiRNA production. We also examined the effect of RTCB knockdown on ANG-induced tiRNA production using *in lysate* ANG digestion method we recently reported [44]. RTCB knockdown increased the amount of tiRNAs generated by *in lysate* ANG digestion (Appendix A). Although ANG treatment-based tiRNA production is dependent on ANG uptake into the cell, this method is based on the cleavage reaction by ANG directly added into the lysate. Therefore, we can exclude the possibility that RTCB knockdown increased tiRNA production by increasing the uptake of ANG.

We further investigated the effect of RTCB knockdown on tiRNA production by endogenous RNases. In RNH1 knockout cells (ΔRNH1 cells), intracellular RNase A superfamily enzymes including ANG are fully activated even under non-stress condition, producing tiRNAs constitutively [27]. RTCB knockdown significantly increased the amount of tiRNAs (Figure 1D and Appendix A), suggesting that the inhibitory effect of the RTCB-LC is independent of the existence of RNH1-RNase complexes. In addition, RTCB knockdown also enhanced sodium arsenite-induced tiRNA production (Appendix A). Taken together, these data clearly showed that RTCB-LC functions as a negative regulator of tiRNA production.

### 2.2. tiRNAs Can Be Repaired to Full Length tRNAs In Vitro

Next, we examined whether full length tRNAs could be generated from tiRNAs by RTCB-mediated ligation in vitro (Figure 2). We used *E. coli* RtcB for our ligation experiments because RtcB is commercially available and is a stand-alone enzyme [45] in contrast to the human RTCB-LC as a multi-subunit complex. We prepared tiRNAs by *in lysate* ANG digestion followed by gel-purification. The incubation of tiRNAs with recombinant RtcB generated ligation products around 75 nt, which is consistent in the length of mature tRNAs (Figure 2A). We further confirmed that these ligation products were derived from tiRNAs by Northern blotting (Figure 2B), suggesting that RtcB has the capability of generating full-length tRNAs from tiRNAs in vitro. It should be noted that two bands were detected by Northern blotting using 5′-tRNA^Ser^-specific probe (position 1–21) in this in vitro experiment. The lower band (indicated by a blue arrowhead) seems shorter than the intact tRNA^Ser^. Because this shorter band was not detected by Northern blotting using 3′-tRNA^Ser^-specific probe (position 50–82), we suspected that it is the chimeric tRNA derived from 5′-tiRNA^Ser^ and non-serine 3′-tiRNA.

To confirm whether RtcB can generate chimeric tRNAs in vitro, we examined the generation of chimeric tRNAs using synthetic tiRNAs (Appendix A). First, we examined the ligation of the mixture of 5′-tiRNA^Ser^, 3′-tiRNA^Ser^ and 3′-tiRNA^Gly^ (Appendix A). When the mixture was incubated with RtcB, 5′-tiRNA^Ser^ was ligated specifically to 3′-tRNA^Ser^, generating the full length (85 nt) tRNA^Ser^. The chimeric tRNA consisting of 5′-tiRNA^Ser^ and 3′-tiRNA^Gly^ was not detected in SYBR Gold staining or Northern blotting (Appendix A). Note that the band detected using the tRNA^Gly-GCC^ (position 37–64) probe was considered as a dimer of 3′-tiRNA^Gly^ (indicated by an asterisk), as it was also detected by SYBR Gold staining of 3′-tiRNA^Gly^ (Appendix A). We also examined the ligation of the mixture of 5′-tiRNA^Gly^, 3′-tiRNA^Gly^ and 3′-tiRNA^Ser^ (Appendix A). In contrast, 5′-tiRNA^Gly^ was nonspecifically ligated to both 3′-tiRNA^Gly^ and 3′-tiRNA^Ser^, generating the chimeric tRNA consisting of 5′-tiRNA^Gly^ and 3′-tiRNA^Ser^ as well as the full length tRNA^Gly^ (Appendix A). These data suggests that RtcB can generate chimeric tRNAs from the mixture of tiRNAs depending on the combination of tiRNAs in in vitro settings. We hypothesized that the selectivity of the ligation was partly endowed by annealing of two tiRNAs in the stem structure. Next, we synthesized 5′-tiRNAs^Ser^ possessing 3 or 6 mutations in stems (acceptor and anticodon stems) that inhibit the annealing to 3′-tiRNA^Ser^ (Appendix A), then investigated the effect of the mutations on ligation efficiency. As a result, as mismatches within the stem structure increased, the amount of ligation products decreased (Appendix A), which suggests that the efficiency of annealing in the stem structure between 5′-tiRNAs and 3′-tiRNAs may contribute to the selectivity of RtcB-mediated ligation.

### 2.3. Chimeric tRNAs Are Not Detected in the Cell

If RTCB-mediated ligation induced substantial amount of chimeric tRNAs in the cell, it could be detrimental to translation because the anticodons of chimeric tRNAs would not correspond to the amino acids charged to their 3′-end. Therefore, we investigated whether chimeric tRNAs were generated after ANG-mediated tRNA cleavage in the cell by next generation sequencing (Figure 3) using the cDNA libraries derived from gel-purified tRNA fractions from ANG treated cells we previously reported [27].

We examined whether the chimeric tRNAs consisting of the 5′-side of tRNA^Gly-GCC-1^ (i.e., 5′-tiRNA^Gly-GCC-1^) were detected or increased by ANG treatment for the reasons below. If chimeric tRNAs existed, they would need to be cleaved into tiRNAs by ANG before RTCB-mediated ligation. Therefore, the chimeric tRNAs derived from abundant tiRNAs are expected to be easily detected. A number of studies reported that 5′-tiRNA^Gly-GCC^ is one of the most abundant tiRNAs [46,47,48,49], suggesting that tRNA^Gly-GCC^ is efficiently cleaved by ANG. Using the cDNA library of tiRNA fractions (20–50 nt) obtained from ANG-treated cells [27], we confirmed the cleavage site of tRNA^Gly-GCC-1^ by ANG (Appendix A). Under basal conditions, various length of fragments derived from tRNA^Gly-GCC-1-1^ were detected (Appendix A). However, in ANG-treated cells, the read counts of 34-nt 5′-tiRNA^Gly-GCC-1-1^ were specifically increased (Appendix A), suggesting that ANG exclusively targets between C and C in the anticodon of tRNA^Gly-GCC-1^, generating the 34-nt 5′-tiRNA^Gly-GCC-1^ and the 40-nt 3′-tiRNA^Gly-GCC-1^ (Figure 3A and Appendix A). According to the flowchart shown in Figure 3B, we investigated whether ANG-treatment induced chimeric tRNAs consisting of 5′-tiRNA^Gly-GCC-1^ and non-Gly 3′-tiRNA (see Materials and Methods for detail). First, we extracted the reads the 5′-side of which were mapped to tRNA^Gly-GCC-1^ genes, then examined whether their 3′- counterparts were mapped to other tRNA genes (Figure 3B). If the 3′-side of the reads were mapped to other tRNA genes, the reads would be considered as chimeric tRNAs. As a result, we could not detect any chimeric tRNAs irrespective of ANG treatment (Figure 3C). It is estimated that production of chimeric tRNA is very rare in the cell even if they could be produced (discussed later).

### 2.4. tiRNA Production Is Enhanced under Oxidative Stress Conditions through RTCB Inhibition

Recently, it has been reported that the RTCB-LC is inhibited under oxidative stress conditions through copper-mediated oxidation [43]. In this report, it was shown that oxidative stress inducers such as hydrogen peroxide (H_2_O_2_) and menadione, inhibited the enzymatic activity of the RTCB-LC, and RTCB inhibition can be detected by the accumulation of pre-tRNA^Tyr^-derived fragment [43]. Therefore, we examined whether tiRNA production was enhanced under two kinds of oxidative stress conditions, H_2_O_2_ treatment (Figure 4A,B) and menadione treatment (Figure 4C,D). First, we confirmed that both H_2_O_2_ (at 1 mM) and menadione (at 100 µM) induced the production of pre-tRNA^Tyr^-derived fragments at 2 h (Appendix A). As sodium arsenite (SA) treatment activates ANG and other RNase A superfamily enzymes in the cell through dissociation of RNH1 [27], we examined whether SA-induced tiRNA production was enhanced by combination with H_2_O_2_ treatment (Figure 4A,B) or menadione treatment (Figure 4C,D). As shown in Figure 4A,B, H_2_O_2_ itself did not induce efficient tiRNA production, suggesting that 2 h exposure to H_2_O_2_ did not induce the efficient dissociation of RNH1 from RNases. However, additional H_2_O_2_ treatment significantly boosted SA-induced tiRNA production (Figure 4A,B and Appendix A). It is worth noting that the additional bands were detected only by the combination of H_2_O_2_ and SA (Figure 4B and Appendix A), suggesting that the combination treatment induced the cleavage at other (than anticodon loop) sites of tRNAs, although the mechanism has yet to be defined. Under the condition, the protein levels of RNH1 did not differ significantly between groups (Appendix A). There were also no marked differences in the mRNA levels of RNASE1, RNASE4 and ANG between groups (Appendix A). These data suggest that the degradation of RNH1 or up-regulation of RNases is not a main determinant of tiRNA production, supporting that the net amount of tiRNAs was mainly determined by both the activation of RNases and the inhibition of the RTCB-LC under this condition. In addition, H_2_O_2_ treatment significantly enhanced tiRNA production in RNH1 knockout (ΔRNH1) cells (Appendix A) similarly to RTCB knockdown.

We also examined whether menadione, another oxidative stress inducer, enhanced SA-induced tiRNA production (Figure 4C,D). Similarly to the previous experiment, the amount of tiRNAs was increased by the combination of menadione and SA compared to SA alone (Figure 4C,D). However, in clear contrast to H_2_O_2_, the treatment with menadione alone induced tiRNA production as much as the combination of SA and menadione did (Figure 4C,D). SA treatment alone did not induce pre-tRNA^Tyr^-derived fragments (Figure 4D), suggesting that SA does not inhibit the RTCB-LC through oxidation under the condition. The amount of pre-tRNA^Tyr^-derived fragments did not differ between menadione alone and menadione + SA, suggesting that SA does not further inhibit the RTCB-LC. These data suggest that menadione may induce not only oxidative stress but also dissociation of RNH1 from RNases.

### 2.5. Inhibition of RTCB by Oxidation Is a Major Regulator of tiRNA Production

Although this RTCB inhibition can be detected as a pre-tRNA^Tyr^-derived fragment, there does not exist a reliable method to evaluate stress-induced RNH1 dissociation. Therefore, we developed the method to visualize how many RNases are activated in response to stress stimuli. We transiently overexpressed the C-terminally His-tagged ANG (ANG-His) variant, and immuno-precipitated ANG-His using anti-His-tag antibody (Figure 5A). Then, we detected the RNH1 that was bound to ANG-His by Western blotting. If ANG is inactivated by RNH1, RNH1 will be co-immunoprecipitated. Therefore, the decrease in ANG-bound RNH1 indicates the activation of ANG (and likely other RNase A superfamily members expressed in the cell). First, we validated the Co-IP method (Figure 5B,C). Using the anti-His-tag antibody and the protein G beads, ANG-His was concentrated in IP fraction (Figure 5B). Under this condition, RNH1 was co-immunoprecipitated with ANG-His, suggesting that at least significant fraction of ANG-His was inactivated by RNH1 (Figure 5B). It should be noted that RTCB was not co-immunoprecipitated (Figure 5B), suggesting that RTCB does not interact with ANG-His or RNH1. We also confirmed that the overexpressed ANG-His was enzymatically functional. As shown in Figure 5C, the overexpression of ANG-His alone did not induce significant tiRNA production, however, once treated with SA, the ANG-His-overexpressed cells generated the greater amount of tiRNAs compared to the control (mock-transfected) cells as previously reported [50]. Using this Co-IP method, we evaluated the dissociation of RNH1 from ANG-His under several stress conditions (Figure 5D). In INPUT fractions, the expression levels of ANG-His did not differ between groups. In IP fractions, the amount of precipitated ANG-His did not change by stresses, suggesting that these stress stimuli did not affect the efficiency of immunoprecipitation of ANG-His. As expected, SA treatment substantially decreased the levels of ANG-bound RNH1, while H_2_O_2_ did not (Figure 5D). As expected in Figure 4C,D, menadione treatment decreased the amount of ANG-bound RNH1, suggesting that menadione induced the dissociation of RNH1 from ANG-His (Figure 5D).

We propose the model of regulation of stress-induced tiRNA production consisting of two factors, (1) a trigger and (2) a booster (Figure 5E). The trigger refers to stress-induced dissociation of RNH1, which results in the activation of intracellular RNase A superfamily enzymes including ANG. We showed that the stress-induced RNH1 dissociation is clearly visualized using our Co-IP system (Figure 5D). The booster refers to the inhibition of the RTCB-LC under oxidative stress conditions. As the RTCB-LC negatively regulates tiRNA production, its inhibition promotes tiRNA production. This RTCB inhibition can be evaluated by the accumulation of pre-tRNA^Tyr^-derived fragments [43]. The amount of pre-tRNA^Tyr^-derived fragments was increased in reverse proportion to the amount of functional RTCB (Appendix A). We confirmed that the net amount of stress-induced tiRNAs can be estimated by these two factors (Figure 5E). In the control (NC), tiRNAs are not produced because both trigger and booster are off. Because H_2_O_2_ treatment activates only the booster, tiRNAs are not produced, as the trigger remains off. SA treatment induces only RNH1 dissociation. Because SA does not inhibit RTCB, tiRNA production is not enhanced. When treated with both H_2_O_2_ and SA, tiRNA production is significantly enhanced because the booster becomes activated in addition to pulling the trigger. Menadione treatment activates not only the booster but also the trigger as predicted in the previous experiments. tiRNA production is boosted by combination of the trigger and the booster.

Although both SA and menadione induced the dissociation of RNH1 from RNases (Figure 5D), the mechanisms of RNH1 dissociation seemed to be different between the two (Figure 5F–H). While menadione-induced tiRNA production was abrogated by the treatment with an antioxidant N-acetyl-L-cysteine (NAC), SA-induced tiRNA production was not inhibited by NAC at all (Figure 5F,G). In addition, although SA is generally considered as an oxidative stress inducer [51], it did not inhibit the RTCB-LC like H_2_O_2_ or menadione (Figure 4C and Figure 5G). The ANG-His Co-IP showed that NAC did not recover SA-induced RNH1 dissociation, while it completely recovered manadione-mediated RNH1 dissociation (Figure 5H), which further supports that the mechanism of SA-induced RNH1 dissociation is independent of oxidative stress. The data also indicates that both RNH1 dissociation and RTCB inhibition induced by menadione were mediated by oxidative stress.

## 3. Discussion

Accumulating evidence suggests that the tiRNA production rates are one of the essential determinants of cell fate including survival during stress. Therefore, it is important to clarify the precise mechanism of regulation of tiRNA production. While sodium arsenite is broadly used to activate intracellular RNases to generate tiRNAs, only 1–2% of mature tRNAs are cleaved into tiRNAs by sodium arsenite treatment [5,52]. Even in RNH1 knockout U2OS cells, in which intracellular RNase A superfamily enzymes are fully activated, the amount of intracellular tiRNAs is small [27]. To respond to rapid environmental changes, cells need to produce tiRNAs efficiently and rapidly. Although it was reported that ANG expression was up-regulated in response to stress [53,54], it takes time to cope with sudden stress by synthesizing RNases. Therefore, we hypothesized that there exist other mechanisms to boost tiRNA production other than up-regulation of RNases. In this study, we showed that the RTCB-LC negatively regulates tiRNA production in the cell (Figure 1). As the RTCB-LC was reported to be inhibited under the oxidative stress conditions [43], we also showed that the inhibition of the RTCB-LC functions as a booster of tiRNA production under oxidative stress conditions (Figure 4). It is worth noting that the inhibition of the RTCB-LC under oxidative stress condition is very rapid process, pre-tRNA^Tyr^-derived fragments were detected at 15 min after exposure of H_2_O_2_ or menadione, leading to the rapid expansion of tiRNA production (Appendix A). This rapid inhibition enables cells to efficiently generate tiRNAs without any transcriptional or translational regulation that are time-consuming.

As the RTCB-LC is an RNA ligase complex, it is natural to consider that the mechanism of tiRNA regulation by the RTCB-LC is to ligate tiRNAs to generate intact tRNAs. As theoretically expected, RtcB was capable generating full length tRNAs from purified tiRNAs in vitro (Figure 2). However, it was suggested that the ligation products contained chimeric tRNAs (Figure 2). Therefore, we investigated whether chimeric tRNAs were generated after ANG-mediated tRNA cleavage in the cell. As a result, we could not detect any chimeric tRNAs containing 5′-tiRNA^Gly-GCC-1^ (Figure 3). We cannot rule out the possibility that chimeric tRNAs were not detected because of the lack of depth in our dataset. However, we can estimate that the generation of chimeric tRNAs is very rare even if they could be produced. The reason of the discrepancy between in vitro and *in cellulo* is considered below. First, to our best knowledge, there is no report about the detection of chimeric tRNAs generated through XBP1 mRNA splicing or pre-tRNA splicing. It was reported that the unspliced XBP1 mRNA forms a conserved bifurcated stem loop structure that contains the intron. The formation of the stem structure through base-pairing of the exons is required for both preventing the separation of exons and facilitating the ligation reaction by a zipper-like mechanism [55]. Although a strong experimental proof is lacking, it is generally proposed that pre-tRNA exons also keep bound after intron removal through the base-pairing in both acceptor and anticodon stems, which prevents exons from being separated. Therefore, a plausible explanation is that cleaved tiRNAs remain bound to their counterparts through their base-pairing in the stem structures for a while. Under this condition, cleaved tiRNAs are not distinct from intact tRNAs except that they possess a nick in the anticodon loop. The RTCB-LC likely ligates these tiRNAs, generating intact tRNAs before separation. On the other hand, the purified tiRNAs used for in vitro ligation experiment are likely to be completely separated and their secondary structures must be lost. Therefore, we reason that RtcB randomly ligate 5′-tiRNAs to closely located 3′-tiRNAs, resulting in the generation of substantial amount of chimeric tRNAs in in vitro setting.

From the results in this study, we propose a model of tiRNA regulation including under oxidative stress conditions (Figure 6). Under normal conditions, intracellular ANG and other RNase A superfamily enzymes are inactivated by RNH1. Stress stimuli such as sodium arsenite activate RNases through RNH1 dissociation, resulting tRNA cleavage in the anticodon. This RNH1 dissociation acts as “the trigger” of tiRNA production. If the RTCB-LC remains intact, it repairs the nick within the anticodon, decreasing the net production of tiRNAs. Although whether there exist mechanisms that facilitate the separation of tiRNAs remains to be elucidated, once separated, tiRNAs should start to function as stress-responsive molecules. Under oxidative stress conditions where the RTCB-LC is inhibited, the efficiency of tRNA repair decreases, resulting in the expansion of functional tiRNAs contributing to cell protection. This oxidation-mediated inhibition of the RTCB-LC acts as “the booster” of tiRNA production (Figure 6). We showed that the net amount of stress-induced tiRNA production can be estimated by assessing two factors, the trigger (RNH1 dissociation) and the booster (RTCB inhibition).

The dissociation of cytoplasmic RNase/RNH1 complex, that is called “the trigger” in this study, has long been proposed as one of the major mechanisms of stress-induced tiRNA production (reviewed in [3,56]). We recently reported that in RNH1 knockout (ΔRNH1) cells, in which intracellular RNase A superfamily enzymes are fully activated, sodium arsenite treatment did not further increase tiRNA production, which suggests that stress-induced tiRNA production was mainly regulated by the dissociation of RNH1 from cytoplasmic RNases [27]. Mechanistically, it was reported that oxidative stress causes the dissociation of RNH1 from bovine RNase A by oxidation of thiol residues in RNH1 in vitro [57]. However, it remains to be clarified whether it is the main mechanism of RNH1 dissociation in the cell. It is also unclear whether there exist other mechanisms inducing RNH1 dissociation. In this study, we developed a simple, but reproducible and effective method to visualize the interaction between RNH1 and ANG-His in the cell (Figure 5). Using this Co-IP system, we clearly showed that not only SA but also menadione induced the dissociation of RNH1 from ANG-His (Figure 5D). We also showed that SA-induced RNH1 dissociation was independent of oxidative stress, while menadione-mediated RNH1 dissociation was oxidative stress-dependent (Figure 5H). These data suggest that there exist both oxidative stress-dependent and -independent mechanisms of stress-induced RNH1 dissociation. Our experimental system may shed light on the molecular mechanisms of RNH1 dissociation in response to stress stimuli.

Our results propose a novel mechanism of reprogramming of RNA metabolism in response to oxidative stress mediated by the RTCB-LC. What is interesting is that tiRNAs are required to evade RTCB-mediated ligation for their function, while XBP1 mRNA exons and pre-tRNA exons must be ligated by the RTCB-LC to become functional molecules. Therefore, once the RTCB-LC is inhibited, cells can enhance tiRNA production simultaneously with repression of pre-tRNA splicing and XBP1 mRNA splicing, which suggests that cells prioritize tiRNA production at the expense of both tRNA biogenesis and ER stress response under oxidative conditions. Although this mechanism seems reasonable for cellular stress response, XBP1 mRNA-mediated ER stress response should be inhibited under the condition. Therefore, it is unclear whether the prioritization of tiRNA production is beneficial for cells under complex stress conditions such as the combination of oxidative stress and ER stress. In addition, the inhibition of the RTCB-LC induces pre-tRNA-derived fragments, some of which were reported to sensitize cells to oxidative stress-induced cell death [58,59]. Further investigation is required to clarify the net effect of oxidative stress-induced RTCB inhibition on cellular stress response.

In summary, the RTCB-LC acts as a negative regulator of tiRNA production under basal condition. Under oxidative stress conditions, oxidation-mediated inhibition of the RTCB-LC boosts stress-induced tiRNA production. This rapid inhibition of the RTCB-LC may act as a cellular mechanism to efficiently trigger the stress response pathway mediated by tiRNAs.

## 4. Materials and Methods

### 4.1. Cell Culture and Treatment

The human osteosarcoma U2OS cells were maintained at 37 °C in a CO_2_ incubator in Dulbecco’s modified Eagle’s medium (DMEM) (08458-45, Nacalai Tesque, Kyoto, Japan) supplemented with 10% fetal bovine serum (35-079-CV, Corning, Corning, NY, USA) and 1% of penicillin-streptomycin mixed solution (Nacalai Tesque, 09367-34). For cellular ANG treatment, cells were incubated with DMEM containing 0.5 µg/mL recombinant human angiogenin (265-AN-250, R&D systems, Minneapolis, MN, USA) for 1h. Sodium arsenite (191-01241, FUJIFILM Wako, Osaka, Japan), hydrogen peroxide (Nacalai Tesque, 18411-25), menadione (Nacalai Tesque, 36405-71), and N-Acetyl-L-cysteine (Nacalai Tesque, 00512-84) were used at 500 µM, 1 mM, 100 µM and 5 mM, respectively. Cells were incubated with these drugs for 2 h unless otherwise indicated.

### 4.2. Knockdown of RTCB

Oligonucleotides encoding shRNAs were synthesized by IDT (Integrated DNA Technologies, Newark, NJ, USA) and subcloned into pLKO.1 vector (Addgene, Watertown, MA, USA) according to the manufacturer’s instructions. The lentiviral particles were produced by co-transfection of the pLKO.1 construct with pMD2G and psPAX2 (Addgene) into Lenti-X 293T cells (Clonetech, Mountain View, CA, USA) using Lipofectamine 3000 (ThermoFisher, Waltham, MA, USA). Cells were infected with lentiviral particles in the presence of 8 μg/mL polybrene (Sigma-Aldrich, St. Louis, MO, USA), then selected one day after viral transduction with 2 μg/mL of puromycin (Sigma-Aldrich). Cells were collected 7 days after transduction unless otherwise indicated. Knockdown efficiency was confirmed by Western blotting using anti-RTCB antibody (sc-393966, Santa Cruz, Dallas, TX, USA). Anti-β-actin antibody (66009-1-Ig, Proteintech, Rosemont, IL, USA) was used as a loading control. The sequences of the shRNA-encoding oligonucleotides were shown in Appendix A.

### 4.3. Western Blotting

Whole cell lysates prepared with RIPA buffer were separated by SDS-polyacrylamide electrophoresis (SDS-PAGE) and transferred onto Immobilon-P PVDF membranes (Millipore, Burlington, MA, USA). Blots were blocked with 5% skim milk in TBS-T at room temperature for 30 min, then incubated with primary antibodies overnight at 4 °C. After washing and 1 h incubation with an HRP-conjugated secondary antibody (115-035-062 or 115-035-144, Jackson ImmunoResearch, West Grove, PA, USA) at room temperature, the signals were detected using a Chemi-Lumi One L (Nacalai Tesque) and the ChemiDoc imaging system (Bio-Rad, Hercules, CA, USA) according to the manufacturer’s instructions.

### 4.4. In lysate Angiogenin Digestion

*In lysate* angiogenin (ANG) digestion method was performed as we previously reported [44]. Briefly, recombinant human ANG was added to the cytoplasmic lysate at the final concentration of 30 nM, followed by incubation at room temperature for 30 min. Total RNAs in the lysate were purified using Sepazol-RNA II Super (Nacalai Tesque, 30487-46).

### 4.5. Northern Blotting

Total RNA was obtained using Sepazol-RNA I Super G (Nacalai Tesque, 09379-55). RNA was run on 10% TBE-urea gels, transferred to positively charged nylon membranes (Hybond N+, Cytiva, Marlborough, MA, USA), then UV cross-linked. The membranes were then hybridized at 40 °C for 2 h with digoxigenin (DIG)-labeled DNA probes in DIG Easy Hyb solution (Roche, Basel, Switzerland). After washing twice with 1× TBS-T, the membranes were blocked in blocking reagent (Roche) at room temperature for 30 min, then probed with alkaline phosphatase-labeled anti-digoxigenin antibody (Roche) for 30 min. Signals were visualized with CDP-Star ready-to-use (Roche) and detected using ChemiDoc imaging system (BioRad) according to the manufacturer’s instructions. Oligonucleotide probes were synthesized by IDT. Digoxigenin-dUTP was added to the 3′-end of the probes using the DIG Oligonucleotide tailing kit (Roche). The sequences of the probes are shown in Appendix A. Densitometry was performed using ImageJ software (NIH).

### 4.6. Ligation of tiRNAs by Bacterial RtcB Ligase

For the ligation of endogenous tiRNAs, tiRNAs were prepared by *in lysate* ANG digestion [44] at 100 nM ANG, followed by gel-purification using ZR small-RNA PAGE Recovery Kit (R1070, Zymo Research, Irvine, CA, USA). The purified tiRNAs were first denatured by heating at 90 °C for 2 min, then mixed with 10x Reaction buffer and RNase inhibitor (Nacalai Tesque, 30260-96) followed by incubation at 37 °C for 10 min for annealing. After mixing with GTP, MnCl_2_ and 15 pmol of RtcB ligase (M0458, New England Biolabs, Ipswich, MA, USA) according to the manufacturer’s instruction, the mixture was incubated at 37 °C for 1 h.

For the ligation of synthetic tiRNAs to examine the generation of chimeric tRNAs, ligation reaction was performed as described above using 40 pmol of 5′-tiRNA and 20 pmol each of two kinds of 3′-tiRNAs. For the ligation of mismatched tiRNAs^Ser^, 20 pmol each of 5′-tiRNA^Ser^ and 3′-tiRNA^Ser^ were incubated with RtcB at 37 °C for 20 min. The sequences of synthetic tiRNA oligos are shown in Appendix A.

### 4.7. Pulldown of 6xHis-Tagged Angiogenin

The full-length CDS of human ANG was PCR amplified so that 6x His tag was added at the C-terminal end. The PCR amplicon was then subcloned into the BamH I site of the pcDNA3.1 vector (ThermoFisher). Two microgram of the construct (pcDNA-ANG-His) was transfected into U2OS cells (5 × 10^5^ cells) in a 6-well dish using Lipofectamine 3000 (ThermoFisher) according to the manufacturer’s protocol for reverse transfection. Forty-eight hours after transfection, cells were lysed with the lysis buffer (0.5% NP-40-containing PBS). After removing the nuclei by centrifugation, the lysate was diluted 10 times with the washing buffer (0.1% NP-40-containing PBS) supplemented with protease inhibitor cocktails (Nacalai Tesque, 25955-11), and incubated with 2.5 µg of anti-6xHistidine monoclonal antibody (FUJIFILM Wako, 014-23221) conjugated to 25 µL of Protein G Sepharose 4 Fast Flow (cytiva, 17-0618-01) at room temperature for 1 h with gentle rotation. After washing the beads twice with the washing buffer, immunoprecipitated proteins were eluted in 1x SDS sample buffer by boiling. The amounts of ANG-His and ANG-bound RNH1 were evaluated by Western blotting using anti-ANG antibody (Santa Cruz, sc-74528) and anti-RNH1 antibody (Proteintech, 10345-1-AP), respectively.

### 4.8. Small RNA-Seq Data Analysis for Chimeric tRNA Generation

The cDNA library of tRNA fractions obtained from ANG-treated U2OS cells (PRJNA770144) (described in detail in [27]) was used in this study. To examine whether ANG-mediated tRNA cleavage induces the generation of chimeric tRNAs, first, both the adapter sequences and the 3′-CCA were trimmed using Trimmomatic v0.39 [60]. Then, the reads were divided into two portions, position 1–34 (the 5′-side) and the other (the 3′-side) using Seqtk-1.3 (r106) (https://github.com/lh3/seqtk, accessed on 29 June 2022). Next, the 5′-side of reads were aligned to hg38 reference genome (obtained from UCSC Genome Browser [61]) using Bowtie 1.0.0 [62] with parameters “-S -k 1 --best -v 2”, then the reads mapped tRNA^Gly-GCC^^-1^ were extracted using seqkit ver 2.2.0 [63]. Finally, the counterparts (the 3′-side) of the extracted reads were examined whether they were aligned to tRNA genes other than tRNA^Gly-GCC^^-1^ using Bowtie. If they were aligned to non-tRNA^Gly-GCC^^-1^ genes, they would be chimeric tRNAs consisting of 5′-tiRNA^Gly-GCC^^-1^ and 3′-tiRNAs derived from non-tRNA^Gly-GCC-1^.

To identify the cleavage site of tRNA^Gly-GCC-1^ by ANG, the cDNA library of tiRNA fractions (20–50 nt) obtained from ANG-treated U2OS cells (PRJNA770135) was used (described in detail in [27]). CCA- and adapter-trimmed reads were aligned to hg38 reference genome using Bowtie as previously reported [27]. The reads mapped to tRNA^Gly-GCC-1-1^ gene were visualized using the Integrated Genome Viewer (IGV) software [64].

### 4.9. Quantitative PCR

Quantitative PCR was performed using the StepOnePlus Real Time PCR System (Applied Biosystems, Foster City, CA, USA) as previously described [27].

### 4.10. Statistical Analyses

Student’s t-test was used to compare mean values between two groups.

## Figures and Tables

**Figure 1 ijms-23-13100-f001:**
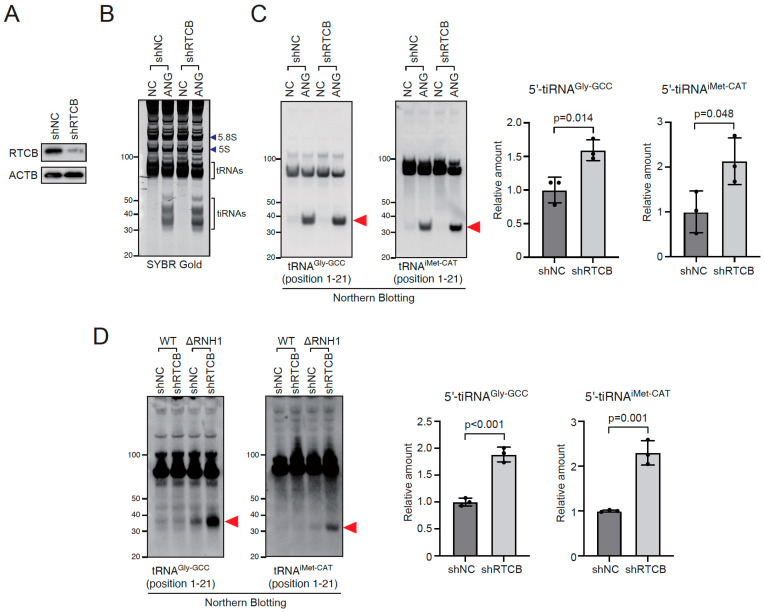
RTCB ligase complex negatively regulates tiRNA production. (**A**) shRNA-mediated knockdown of RTCB. (**B**,**C**) RTCB knockdown enhances ANG-mediated tiRNA production. U2OS cells were incubated with 0.5 µg/mL ANG for 2 h. (**B**) SYBR Gold staining and (**C**) Northern blotting for tRNA^Gly-GCC^ and tRNA^iMet-CAT^. Relative amounts of 5′-tiRNAs calculated by densitometry are also shown (n = 3 each). (**D**) RTCB knockdown boosts constitutively generated tiRNAs in RNH1 knockout cells. Northern blotting for tRNA^Gly-GCC^ and tRNA^iMet-CAT^, and relative amounts of 5′-tiRNAs are shown (n = 3 each). Red arrowheads indicate tiRNAs.

**Figure 2 ijms-23-13100-f002:**
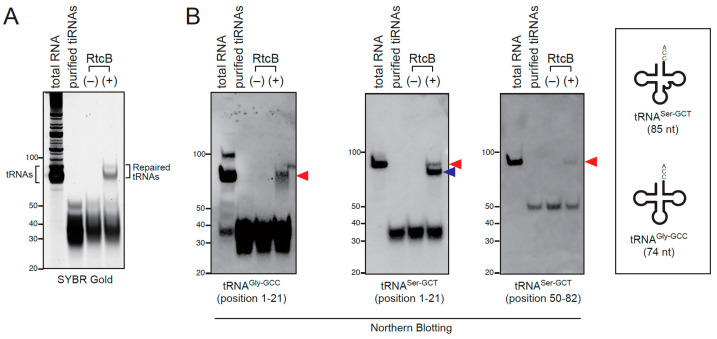
tiRNAs can be repaired to full-length tRNAs by RtcB in vitro. Gel-purified tiRNAs were incubated with bacterial RtcB. (**A**) SYBR Gold staining and (**B**) Northern blotting for tRNA^Gly-GCC^ and tRNA^Ser-GCT^. The lengths of both tRNA^Gly-GCC^ and tRNA^Ser-GCT^ are also shown. tRNA^Ser-GCT^ is longer than tRNA^Gly-GCC^ because of its long variable loop. Ligation products are indicated by arrowheads. Red arrowheads indicate the ligation products that showed the same mobility as that of the corresponding mature tRNA in total RNA. Note that two bands were detected by the probe for 5′-side (position 1–21) of tRNA^Ser-GCT^, in contrast to a single band detected by the probe targeting 3′-side (position 50–82) of tRNA^Ser-GCT^, which suggests that this shorter band (indicated by a blue arrowhead) consists of 5′-tiRNA^Ser-GCT^ and 3′-tiRNAs derived from non-serine tRNAs.

**Figure 3 ijms-23-13100-f003:**
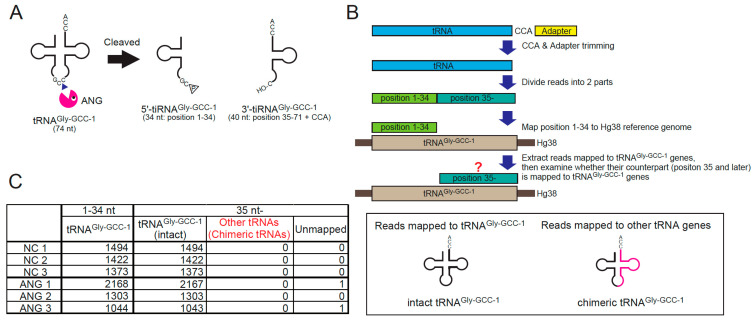
Chimeric tRNAs are not detected *in cellulo*. (**A**) ANG-mediated cleavage site in tRNA^Gly-GCC-1^. ANG cleaves between C and C in the anticodon of tRNA^Gly-GCC-1^, generating 34-nt 5′-tiRNA^Gly-GCC-1^ and 40-nt 3′-tiRNA^Gly-GCC-1^. (**B**) Flowchart of the analysis to detect chimeric tRNAs comprising 5′-tRNA^Gly-GCC-1^. For the detail, see Materials and Methods. (**C**) ANG treatment does not induce chimeric tRNA production in the cell.

**Figure 4 ijms-23-13100-f004:**
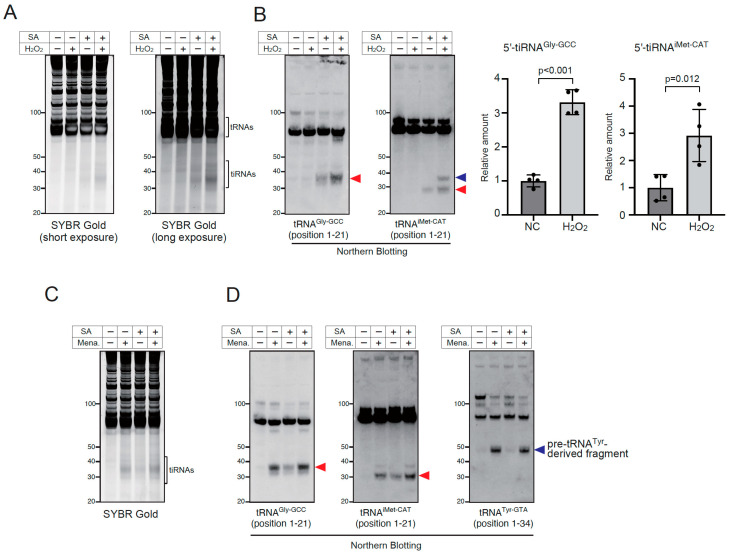
Oxidative stress boosts tiRNA production through inhibition of the RTCB ligase complex. (**A**,**B**) H_2_O_2_ treatment boosts sodium arsenite-induced tiRNA production. U2OS cells were treated with 500 µM of sodium arsenite and/or 1 mM of H_2_O_2_ for 2 h. (**A**) SYBR Gold staining and (**B**) Northern blotting for tRNA^Gly-GCC^ and tRNA^iMet-CAT^. Canonical 5′-tiRNAs are indicated by arrowheads, while a non-canonical 5′-tiRNA^iMet-CAT^ is indicated by a blue arrowhead. The amounts of 5′-tiRNAs calculated by densitometry are also shown (n = 4 each). (**C**,**D**) Effect of the combination between menadione (at 100 µM) and sodium arsenite (at 500 µM) on tiRNA production. (**C**) SYBR Gold and (**D**) Northern blotting for tRNA^Gly-GCC^, tRNA^iMet-CAT^ and tRNA^Tyr-GTA^. tiRNAs are indicated by red arrowheads. SA, sodium arsenite; Mena., menadione.

**Figure 5 ijms-23-13100-f005:**
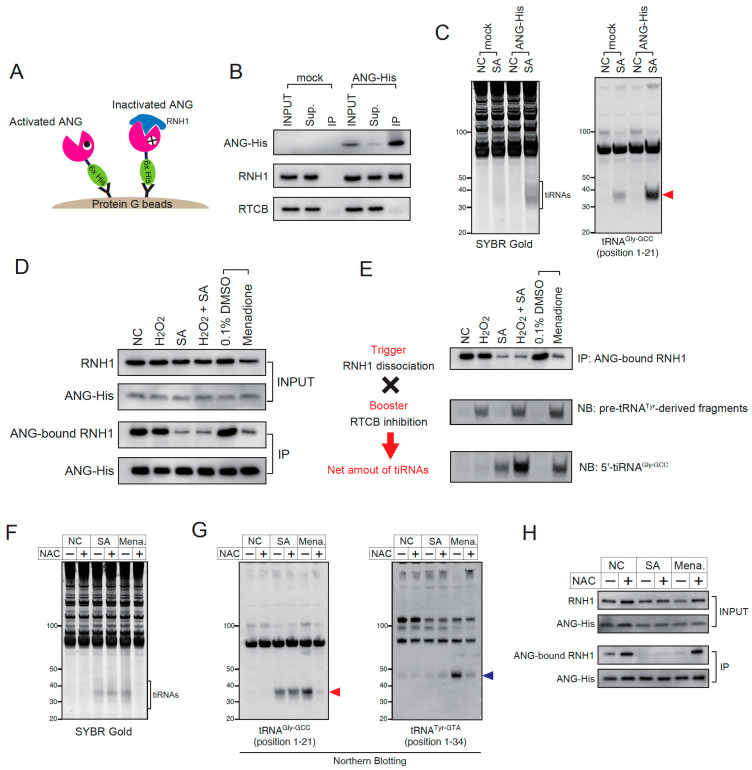
Stress-induced tiRNA production is regulated by both RNH1 dissociation and RTCB inhibition. (**A**–**D**) Evaluation of stress-induced RNH1 dissociation from angiogenin. (**A**) Schematic model of ANG-His pulldown. When ANG was inactivated by RNH1, RNH1 is co-immunoprecipitated with ANG-His. Under the condition where ANG is activated, RNH1 is not co-immunoprecipitated because it dissociates from ANG. (**B**) Efficiency of immunoprecipitation. Overexpressed ANG-His was concentrated by immunoprecipitation. Under non-stress condition, RNH1 is co-immunoprecipitated with ANG-His. Note that RTCB was not co-immunoprecipitated with ANG-His, suggesting that RTCB does not interact with ANG or RNH1. (**C**) Overexpressed ANG-His is enzymatically functional. Overexpression of ANG-His enhanced sodium arsenite-induced tiRNA production. 5′-tiRNA^Gly-GCC^ is indicated by a red arrowhead. Note that ANG-His overexpression does not induce tiRNA production under non-stress conditions, suggesting that ANG-His is completely inhibited by RNH1 under basal conditions. (**D**) ANG-His pulldown under various stress stimuli. (**E**) The net amount of tiRNAs is regulated by both RNH1 dissociation and RTCB inhibition. RNH1 dissociation (trigger), RTCB inhibition (booster) and the net amounts of tiRNAs are evaluated by ANG-bound RNH1, pre-tRNA^Tyr^-derived fragment and 5′-tiRNA^Gly-GCC^, respectively. (**F**–**H**) The effect of N-acetyl-L-cysteine treatment (at 5 mM) on tiRNA production. (**F**) SYBR Gold staining and (**G**) Northern blotting for tRNA^Gly-GCC^ and tRNA^Tyr-GTA^. 5′-tiRNA^Gly-GCC^ is indicated by a red arrowhead, pre-tRNA^Tyr-GTA^-derived fragments generated due to RTCB inhibition is indicated by a blue arrowhead. (**H**) ANG-His pulldown. Note that N-acetyl-L-cysteine treatment did not prevent RNH1 dissociation, suggesting that sodium arsenite-induced RNH1 dissociation is independent of oxidative stress. NC, negative control; SA, sodium arsenite; Mena., menadione; NAC, N-acetyl-L-cysteine.

**Figure 6 ijms-23-13100-f006:**
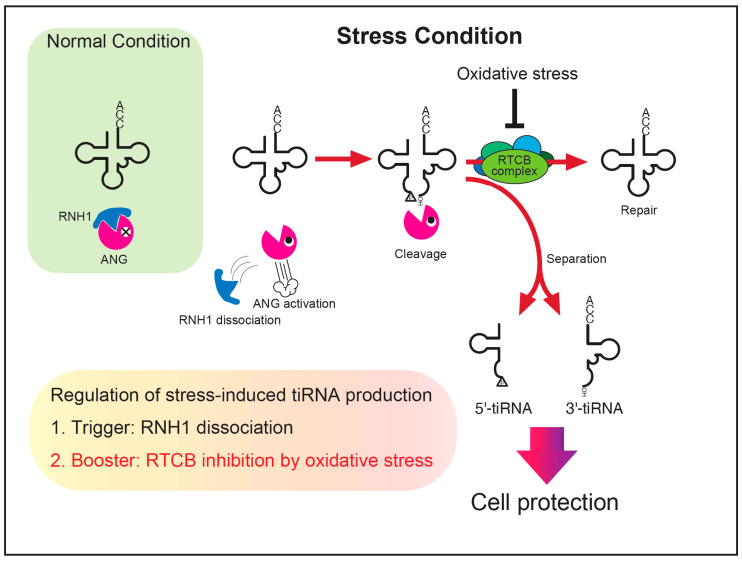
A proposed model of the regulation of stress-induced tiRNA production. Under normal conditions, ANG (and other RNase A superfamily enzymes) is inhibited by RNH1. Once cells are exposed to stress, the RNases become activated through RNH1 dissociation, cleaving tRNAs into tiRNAs. A part of tiRNAs is repaired to intact tRNAs before separation. Under the condition where oxidative stress coexists, the repair of tRNAs is repressed due to inhibition of the RTCB ligase complex, resulting in increased levels of separated tiRNAs that contribute to cell protection against adverse conditions. We propose that the net amount of tiRNA production is estimated by two factors, a trigger (RNH1 dissociation) and a booster (RTCB inhibition).

## Data Availability

Not applicable.

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
