# Peer review of "RTCB Complex Regulates Stress-Induced tRNA Cleavage"

_ijms, 2022, doi:10.3390/ijms232113100_

Round 1

Reviewer 1 Report

This manuscript reports a new finding of tiRNA booster, the RTCB ligase. Convincing evidence is presented that RTCB complex regulates the final production of tiRNA. The findings are significant, and the model that a trigger (RNH1 dissociation) and a booster (RTCB inactivation) collaboratively regulate ANG-mediated tiRNA is novel.

Some minor comments:

Page 3, line 147. I suggest change “….independent of stress condition” to “…. Independent of existence of RNH1-RNase complex” to be more precisely reflecting the data.

Page 6, lines 218 to 223. Figure S6 should be Figure S7

Figure 3C. How do you explain the number of reads of 1-34 nt and 35 nt- are the same? If tRNA-Gly-GCC-1 is cleaved at position 34, wouldn’t you expect more reads of 1-34 nt than of 35 nt-?

Page 7, line 255. Figure S7B should be Figure S9B.

Page 7, lines 285-286. The expression level of ANG was not examined so the possibility that menadione may enhance ANG expression cannot be excluded at present. It is possible that enhanced tiRNA production might be caused by ANG upregulation rather than dissociation of RNN1. The conclusion “that menadione induced not only oxidative stress but also dissociation of RNH1 from RNases” needs to be revised.

Author Response

Reviewer's comments

Response:

We thank this reviewer for the careful and supportive evaluation of our work!

Reviewer 1.

This manuscript reports a new finding of tiRNA booster, the RTCB ligase. Convincing evidence is presented that RTCB complex regulates the final production of tiRNA. The findings are significant, and the model that a trigger (RNH1 dissociation) and a booster (RTCB inactivation) collaboratively regulate ANG-mediated tiRNA is novel.

Some minor comments:

Page 3, line 147. I suggest change “….independent of stress condition” to “…. Independent of existence of RNH1-RNase complex” to be more precisely reflecting the data.

We agree with your suggestion. We changed "independent of stress condition" into "independent of the existence of RNH1-RNase complexes" in the manuscript.

Page 6, lines 218 to 223. Figure S6 should be Figure S7

We corrected the mistakes.

Figure 3C. How do you explain the number of reads of 1-34 nt and 35 nt- are the same? If tRNA-Gly-GCC-1 is cleaved at position 34, wouldn’t you expect more reads of 1-34 nt than of 35 nt-?

Thank you for your comment. In this experiment, we used a cDNA library derived from full length tRNA fraction (about 50-110 nt) that does not contain tiRNAs (30-50 nt). So tiRNAs were not included in the library. The object of this experiment is to detect the ligation products that consist of 5'-side (position 1-34) of tRNA-Gly-GCC-1 (5'-tiRNA-Gly-GCC-1) and 3'-side of other non-Gly tiRNAs generated by RTCB-mediated ligation followed by ANG-mediated cleavage. Because we divided each read (in tRNA fraction) into position 1-34 and the later, the number of reads of 1-34 must be the same as that of 35 nt-. Because every read has its specific ID, we were able to prepare the pair of 1-34 and 35nt- reads derived from the same reads.

Page 7, line 255. Figure S7B should be Figure S9B.

We corrected the mistake. Thank you for pointing it out.

Page 7, lines 285-286. The expression level of ANG was not examined so the possibility that menadione may enhance ANG expression cannot be excluded at present. It is possible that enhanced tiRNA production might be caused by ANG upregulation rather than dissociation of RNN1. The conclusion “that menadione induced not only oxidative stress but also dissociation of RNH1 from RNases” needs to be revised.

Response:

It is true that we did not examine the expression levels of ANG in this experiment. Therefore, we cannot rule out the possibility at this point that there were other factors affecting tiRNA production like ANG upregulation as suggested. We changed "menadione induced..." into "menadione may induce...". But in the following experiment, we confirmed that menadione inhibited the activity of RTCB (by detecting pre-tRNATyr-derived fragments) and that menadione induced the dissociation of RNH1 from ANG-His using our co-IP system (Figure 5H). Therefore, we believe that we can say that menadione can induce RNH1 dissociation as well as oxidative stress although there may be other factors that affect tiRNA production at the point of Figure 5. Thank you for your suggestion.

Reviewer 2 Report

In recent years, many studies have shown that tRNA-derived small RNAs can influence many physiological and pathological processes. Investigation of the molecular mechanisms of tiRNA production is important to understand how tiRNA are involved in these biological processes. According to current knowledge, tiRNAs are produced by RNase-mediated cleavage, and mainly in ANG-dependent manner. Existing studies mainly focused on the factors that involved in ANG-mediated cleavage. In this study, Ivanov and colleagues raised a new model, that tiRNA production consists of both a trigger and a booster process. And the authors identified that RTCB complex, a known RNA ligase complex involved in tRNA splicing, can negatively regulate tiRNA production. Further, they thoroughly studied the regulatory mechanisms of RTCB complex involved in the booster process using both in vitro biochemical and in vivo assays. Their novel findings provide a new direction to understand the regulatory mechanisms of tiRNA production. The manuscript is well-written and the results are nicely organized and hence I strongly recommend the publication at International Journal of Molecular Sciences. I only have a minor point as listed.

1.    Under several oxidative stress conditions, the inhibition of RTCB was only evaluated by the accumulation of pre-tRNAtyr-derived fragments. The author can also determine the RTCB protein inhibition level by western blot.

Author Response

Reviewer's comments

Response:

We thank this reviewer for such an enthusiastic evaluation of our work!

Reviewer 2.

In recent years, many studies have shown that tRNA-derived small RNAs can influence many physiological and pathological processes. Investigation of the molecular mechanisms of tiRNA production is important to understand how tiRNA are involved in these biological processes. According to current knowledge, tiRNAs are produced by RNase-mediated cleavage, and mainly in ANG-dependent manner. Existing studies mainly focused on the factors that involved in ANG-mediated cleavage. In this study, Ivanov and colleagues raised a new model, that tiRNA production consists of both a trigger and a booster process. And the authors identified that RTCB complex, a known RNA ligase complex involved in tRNA splicing, can negatively regulate tiRNA production. Further, they thoroughly studied the regulatory mechanisms of RTCB complex involved in the booster process using both in vitro biochemical and in vivo assays. Their novel findings provide a new direction to understand the regulatory mechanisms of tiRNA production. The manuscript is well-written and the results are nicely organized and hence I strongly recommend the publication at International Journal of Molecular Sciences. I only have a minor point as listed. 

  1. Under several oxidative stress conditions, the inhibition of RTCB was only evaluated by the accumulation of pre-tRNAtyr-derived fragments. The author can also determine the RTCB protein inhibition level by western blot.

Thank you for your comment. We could determine RTCB protein levels by western blot as you suggested. But in Asanovic's Mol Cell paper (ref. 43), they clearly showed that oxidizing agents (H2O2 and menadione) inhibit the enzymatic activity of RTCB through oxidation of SH residue of RTCB. Therefore, we skipped the western blotting experiment as we thought that the effect of oxidative stress on RTCB degradation should be much smaller than on the enzymatic activity of RTCB.

Reviewer 3 Report

Dear Authors, 

It a good contribution

Congratulations!

Author Response

Reviewer's comments

Reviewer 3.

Dear Authors, 

It a good contribution

Congratulations!

Response:

We thank this reviewer for such an enthusiastic evaluation of our work!